# A 14,000-year-old genome sheds light on the evolution and extinction of a Pleistocene vulture

Per G. P. Ericson [1✉], Martin Irestedt[1], Dario Zuccon[2], Petter Larsson[1,3], Jean-Luc Tison[4], Steven D. Emslie[5], Anders Götherström[3,6], Julian P. Hume[7], Lars Werdelin[8] & Yanhua Qu [1,9✉]

The New World Vulture [Coragyps] occidentalis (L. Miller, 1909) is one of many species that were extinct by the end of the Pleistocene. To understand its evolutionary history we sequenced the genome of a 14,000 year old [Coragyps] occidentalis found associated with megaherbivores in the Peruvian Andes. occidentalis has been viewed as the ancestor, or possibly sister, to the extant Black Vulture Coragyps atratus, but genomic data shows occidentalis to be deeply nested within the South American clade of atratus. Coragyps atratus inhabits lowlands, but the fossil record indicates that occidentalis mostly occupied high elevations. Our results suggest that occidentalis evolved from a population of atratus in southwestern South America that colonized the High Andes 300 to 400 kya. The morphological and morphometric differences between occidentalis and atratus may thus be explained by ecological diversification following from the natural selection imposed by this new and extreme, high elevation environment. The sudden evolution of a population with significantly larger body size and different anatomical proportions than atratus thus constitutes an example of punctuated evolution.

[1] Department of Bioinformatics and Genetics, Swedish Museum of Natural History, P.O. Box 50007, SE-10405 Stockholm, Sweden. [2] Institut de Systématique, Evolution, Biodiversité (ISYEB), UMR7205 CNRS MNHN UPMC EPHE Sorbonne Université, Muséum National d'Histoire Naturelle, 75005 Paris, France. [3] Centre for Palaeogenetics, Stockholm, Sweden. [4] Department of Laboratory Medicine, Örebro University Hospital; Södra Grev Rosengatan, SE-70185 Örebro, Sweden. [5] Department of Biology and Marine Biology, University of North Carolina; Wilmington, 601S. College Road, Wilmington, NC 28403, USA. [6] Department of Archaeology and Classical Studies, Stockholm University, SE-10691 Stockholm, Sweden. [7] Bird Group, Department of Life Sciences, Natural History Museum, Akeman St, Tring, Herts, UK. [8] Department of Palaeobiology, Swedish Museum of Natural History, P.O. Box 50007, SE-10405 Stockholm, Sweden. [9] Key Laboratory of Zoological Systematics and Evolution, Institute of Zoology, Chinese Academy of Sciences, Beijing 100101, China. ✉email: per.ericson@nrm.se; quyh@ioz.ac.cn

Paleogenomics, i.e., the genome-scale sequencing of specimens that are many thousands of years old, has facilitated the study of evolutionary processes at the genetic level in populations that are long gone and has become a valuable complement to the fossil record[1–3]. Through genetic data, we can clarify phylogenetic relationships of extinct species, test biogeographic and evolutionary hypotheses, and study factors that have led to the extinction of ancient populations and species. Much paleogenomic work of the last decades (e.g., refs. [4–6]) has focused on the dynamics and demography of the many species and populations that went extinct at the end of the Pleistocene as a consequence of drastic environmental change and increased hunting pressure[7–9]. These studies have benefitted from the fact that many extinctions took place in temperate regions where the chance of finding fossils with preserved DNA is particularly good.

The Pliocene and Pleistocene evolution of large herbivores megafauna, such as proboscideans, edentates, camelids, and horses, sparked the co-evolution of various specialized predators, scavengers, parasites, etc.[10,11], many of which went extinct along with the megaherbivores at the end of the Pleistocene. In the Americas alone, this was the fate of more than thirty species of mammalian and avian predators and scavengers[12,13]. For example, among the New World Vultures (Cathartidae), only 5 out of 11 genera survived into the Holocene[11,14]. Herein, we combine genetic and paleontological data to gain insights into the evolution and extinction of one of these fossil birds, the New World Vulture [*Coragyps*] *occidentalis* (Miller 1909). We argue that this bird co-evolved with the mammalian megaherbivores during the ice age and that its extinction at the end of the Pleistocene coincides with the extinction of these large herbivores.

## Results and discussion

**[*Coragyps*] *occidentalis*-morphological identification, geographic distribution, and paleoecology**. Several skeletal elements of a single individual vulture were collected during paleontological excavations in 1904–1905 in the cave Casa del Diablo, c. 50 km northwest of Lake Titicaca in Peru[15,16] (Supplementary Fig. 1 and Note 1). The specimens are dated to 13,990 ± 110 cal yr BP (Supplementary Table 1). Based on morphology, they are identified as of the extinct vulture [*Coragyps*] *occidentalis* (Fig. 1a and Supplementary Notes 2, 3). This provides the first South American record of a bird previously only known from Late Pleistocene localities in Mexico and USA[17–19] (Fig. 1b and Supplementary Table 2).

In overall appearance, the skeleton of *occidentalis* resembles that of the Black Vulture *Coragyps atratus*, which is common in the lowlands of the Americas, but differs in size, with *occidentalis* being more than 10% larger than *atratus* in many measurements[18]. There are also differences in body and skull proportions, and, to some degree, in skeletal morphology (Fig. 1c, Supplementary Table 3, and Note 4). In skull proportions, *occidentalis* differs from *atratus* in having relatively longer premaxillaries and nares, greater width at the frontonasal hinge, less deep bill, wider and somewhat higher brain case, relatively shorter and stouter limb bones (tarsometatarsus and tibiotarsus), and longer wings[20,21]. Other morphological differences in the skull include, e.g., that the brain case in *occidentalis* being more inflated immediately anterior to the supraoccipital area than in *atratus* and that the supraorbital edges are more excavated posteriorly. Furthermore, the foramen magnum is relatively larger and more compressed vertically in *occidentalis*, the lachrymals having a small opening in the proximal anterolateral surface (instead of a large foramen in *atratus*), and the bill is less hooked[20]. These differences have facilitated taxonomic identification of all but a handful of the *Coragyps* fossils at the 58 sites from where they are reported (Supplementary Table 2).

The precise systematic relationship between *atratus* and *occidentalis* is unknown. It is often assumed that *occidentalis* is the direct ancestor of *atratus*[18,20,22], but the fossil record contradicts this (Supplementary Note 5). The oldest *atratus* fossils date to 1.9–1.6 mya (Early Pleistocene), whereas the oldest confidently identified *occidentalis* fossil is between 0.33 and 0.25 mya (Middle Pleistocene) (Fig. 1d, Supplementary Table 2, and Note 5). It is also clear that the small-sized *atratus* and the large-sized *occidentalis* co-existed in the Late Pleistocene in both North and South America, but seem not to have occupied the same ecological niche. In North America, nearly all certain finds of *occidentalis* are from the southwestern part of the continent, while all confidently identified *atratus* fossils are from the southeastern part (Fig. 1b).

Although this distribution may just suggest a geographic division of a single species into a western and an eastern population, there are strong indications that it instead reflects considerable taxon-specific differences in habitat preferences and adaptation. Today, *atratus* mostly occurs in the lowlands[23] and the fossil record reflects this well; the average elevation for twenty North American sites yielding *atratus* fossils is 42 m a.s.l. (no site is situated higher than 283 m a.s.l.) (Fig. 1d and Supplementary Table 2). In contrast, most *occidentalis* finds are from sites situated in arid mountain regions; the average elevation for the 28 *occidentalis* sites in North America is 1083 m a.s.l. (the highest site is 2280 m a.s.l.). A similar elevational difference is observed also in South America; the single site with *occidentalis* (Peru) is situated at 3819 m a.s.l., while the four Late Pleistocene sites with *atratus* (in Argentina, Ecuador, Peru, and Venezuela) are in the lowlands, at elevations between 25 and 115 m a.s.l. (Fig. 1d and Supplementary Table 2). Although fossils of *occidentalis* are found in arid mountain regions, they are sometimes also found at lowland sites (e.g., Florida, California). However, these lowland locations are mostly places providing extreme feeding opportunities, for example high densities of dead Pleistocene megaherbivores (e.g., in the tar pits in California), accumulations of migrating salmon (at the 5-Mile Rapids in Oregon), and an abundance of both terrestrial and marine carcasses (in the coastal plain in Florida).

**Paleogenomic perspectives on [*Coragyps*] *occidentalis*.** In order to ascertain the generic affinity of *occidentalis*, we extracted DNA from the Casa del Diablo specimens and used whole-genome resequencing to generate a paleogenome of *occidentalis* with a mapping coverage of 1.25× (see Material and Methods and Supplementary Methods 1). We also de novo sequenced and assembled a genome of the extant Black Vulture *Coragyps atratus*. We included the *occidentalis* paleogenome and the *Coragyps atratus* genome in a phylogenomic analysis, including ten species from the clade Accipitrimorphae (including three species of New World Vultures) rooted with two passerine birds and domestic fowl. In the phylogenomic tree, based on 892 exonic regions (Fig. 2a and Supplementary Note 3), *occidentalis* groups with *Coragyps atratus*, the only extant representative of the genus *Coragyps*. The phylogenetic relationships recovered agree with previously published results[24–28] and support the established, morphology-based, generic allocation of *occidentalis*.

To explore the evolution of *occidentalis* in relation to *atratus*, we implemented a principal component analysis (PCA) on genotype likelihoods of genome-wide SNPs from the *occidentalis* fossil and 52 individuals of *atratus* representing its entire geographic range. The PCA plot shows no geographic structuring along the PC1 axis, but the North and Central American *atratus* individuals fall separate from the South American ones along PC2 (Fig. 2b). The *occidentalis* sample falls with the South American

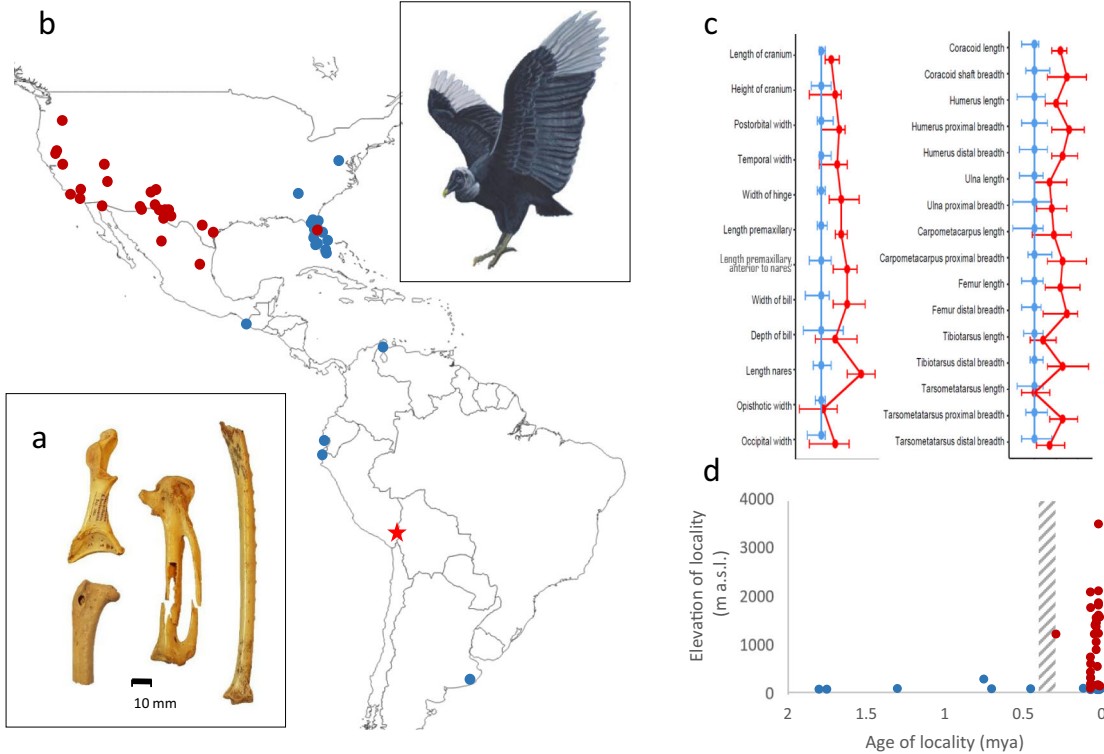

**Fig. 1 The first finds of the extinct [*Coragyps*] *occidentalis* in South America and characterization of all known *Coragyps* fossils. a** Fossils identified as [*Coragyps*] *occidentalis* collected in the cave Casa del Diablo situated at c. 3800 m a.s.l. in the Altiplano of southern Peru (marked with a red star on the map). **b** Localities for fossil *Coragyps* specimens were identified as either *atratus* (blue) or *occidentalis* (red). **c** The ratio diagrams illustrate that *occidentalis* (red) and *atratus* (blue) have marked differences in their body proportions. Skull measurements (left) are from Fisher (1944) and postcranial measurements from Howard (1968) (for data, see S3 Table in S1 Appendix). The whiskers around the mean value indicate the minimum and maximum values observed. If *occidentalis* would be an upscaled version of atratus, the red (*occidentalis*) and blue (*atratus*) lines in the ratio diagrams would be parallel and shifted sideways in relation to the amount of difference in their average sizes. **d** Elevational and temporal distributions of localities yielding fossils of *Coragyps atratus* (blue) and [*Coragyps*] *occidentalis* (red). The gray hatched area indicates the estimated time of the split between *atratus* and *occidentalis* based on molecular data herein. A reconstruction of [*Coragyps*] *occidentalis* by Julian P. Hume is inserted into the map.

individuals, suggesting it is closely related to the extant South American *atratus*. The population genetic structure analysis shows that K = 3 is best supported for these 53 individuals, and the admixture proportions of the fossil are most similar to *atratus* individuals from southwestern South America (Supplementary Fig. 2).

To further clarify the geographic origin of *occidentalis*, we conducted a phylogeographic analysis of 1179 nuclear introns (totaling 586 kb) obtained from 42 recent *atratus* individuals (collected throughout the geographical distribution of *atratus*) and the Casa del Diablo specimen of *occidentalis*, using Bayesian inference. The phylogeny was rooted with one of the samples from Panama (CA15) based on a previous analysis of a smaller number of individuals with higher coverage. The higher coverage allowed us to include exonic regions for which also *Cathartes aura*[29], a representative of the sister clade to *Coragyps*[30], could be aligned (Supplementary Fig. 3). In the population-level phylogeny, the *Coragyps* samples show a distinct geographic clustering (Fig. 2c). From this it can be inferred that the ancestral area of *Coragyps atratus* with 50% probability is in the Panama region, compared to a 25% probability each for a North American or a South American origin (Fig. 2d). If the species evolved in the Panamanian biogeographical area (that includes Central America and northern South America), it may then have spread both north to the Nearctic region via the Mexican biogeographical area, and south to the Amazonian and, subsequently, the Patagonian-Andean biogeographical areas. Such a dispersal scenario also seems more logical than what would follow from

an origin of the species in either North or South America. The temporal and geographic distributions of the *atratus* fossils provide, however, no information on the timing of the dispersals or what routes they followed. Almost all of the oldest finds of *atratus* have been made in Florida (Fig. 1d), which most likely can be explained by differences in the preservation and paleontological activity across its distribution area. The relationships observed in the phylogeographic tree agree well with the assumed geographic distributions of the recognized subspecies of *atratus*[31]. The analyses also suggest a South American origin of *occidentalis* as it groups with two individuals from coastal Chile with strong support (posterior probability >0.98) (Fig. 2c). These two Chilean individuals of *atratus*, together with *occidentalis*, are in turn part of a larger and likewise strongly supported (posterior probability >0.99) clade consisting of individuals from other regions in Chile, Argentina, Paraguay, and southernmost Brazil. The three different methods used to date the split between *occidentalis* and the nearby *atratus* population congruently estimate that this happened c. 300 to 400 kya (Fig. 3a, Supplementary Table 4, and Methods 2). This is in agreement with the paleontological record, where the oldest confidently identified *occidentalis* fossil dates to between 0.33 and 0.25 mya (Middle Pleistocene) (Supplementary Table 2).

Thus, the phylogeographic analysis suggests that the large-sized *occidentalis* evolved from a small-sized *atratus* population living in southwestern South America in the Middle Pleistocene. This is remarkable not only because a population of *atratus* is colonizing a new geographic area, but it also represents a major shift in

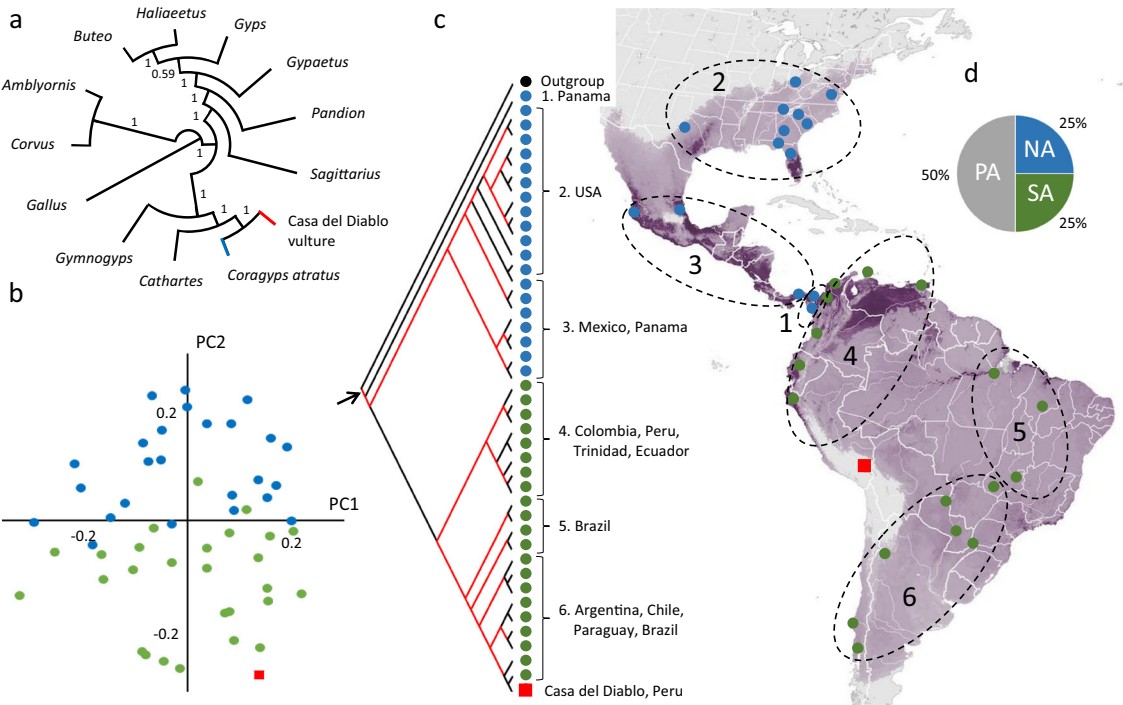

**Fig. 2 Higher-level relationships of the genus *Coragyps*, and phylogenomic structure in *Coragyps atratus*. a** Higher-level phylogeny based on 892 exonic alignments shows that *occidentalis* (red branch) is most closely related to the genus *Coragyps* (blue branch). **b** Principal component analysis of genotype likelihoods estimated from *occidentalis* and 52 individuals of *atratus* representing its entire geographic distribution. The cluster of *atratus* individuals from North and Central America (blue dots) shows only limited overlap with the cluster of South American individuals (green dots) along the PC2 axis. It is also clear that *occidentalis* (red square) is closest to the South American cluster. **c** Phylogenomic analysis of 42 individuals of *Coragyps atratus* and *occidentalis* (red square), rooted with the outgroup, *Cathartes aura* (black dot) based on 1179 intronic alignments (totaling 586 kbp). Red branches have a posterior probability >0.90. The map shows the distribution and abundance of *Coragyps atratus* (from https://science.ebird.org/ based on data in ref. [92]). **d** Inferred ancestral areas for the basal node in Fig. 2c (marked with an arrow; PA Panama, NA North America, and SA = South America).

environmental utilization, which sparked the evolution of adaptations associated with life at high elevations. The harsh conditions in this environment put considerable evolutionary stress on all species living there, and the vulture population should have been subject to strong selective pressure. An increase in body size is a well-documented adaptation to colder environments (ref. [32] ecogeographical rule, Supplementary Note 6) and it is reasonable to assume that the larger size of *occidentalis* compared with *atratus* reflects this. The relatively stouter leg bones of *occidentalis*[18] may also be an adaptation to colder environments (ref.[33] ecogeographical rule). Altogether, *occidentalis* was an evolutionarily distinct form that most likely was adapted to the high elevations it occupied.

The current geographic distribution of *atratus*, along with its fossil record, suggests that it is restricted to lowlands throughout the Americas and has been so for the last 2.0–1.5 mya, or longer. Unquestionably, the colonization of the extreme High Andes and subsequent adaptation to the conditions there represents an important evolutionary step for individuals with ancestry in a lowland *atratus* population. After this population colonized the highlands, it presumably followed its own evolutionary trajectory. The fossil record indicates that *occidentalis* soon began to spread northward to reach the mountainous regions of western North America. Presumably, it dispersed along the Andes and the mountain ridges of Central America, where the lowlands were populated by *atratus*, and there is nothing to suggest otherwise than that *atratus* and *occidentalis* maintained their distinctly different ecological niches.

**Evolutionary history of [*Coragyps*] *occidentalis*.** The evolution and establishment of a viable high-elevation population deriving from *Coragyps atratus*, which normally is restricted to the lowlands, raises several questions. What triggered the initial colonization of the novel environments in the Andes; how could *occidentalis* spread as far north as the Rocky Mountains; and why did it go extinct by the end of Pleistocene after having survived for many hundreds of thousands of years? The fossil record shows that *occidentalis* was closely associated with the Late Pleistocene megaherbivores in North America. This is also true for the Peruvian cave Casa Del Diablo, the only known locality in South America, where *occidentalis* was collected together with remains of camelids, deer, sloths, ground sloths, and horses[15,16]. These and other mammalian megaherbivores (e.g., proboscideans) were widely distributed in the Americas in the Late Pleistocene, and vultures and other scavengers thrived from these. *Coragyps atratus* is associated with megaherbivores at lowland sites from Florida to Peru, and it is likely that the colonization of the highlands was sparked by the occurrence of large mammals in the Andes. Once adapted to this environment, the new taxon *occidentalis* dispersed to other high-elevation regions housing megaherbivores. *occidentalis* thus became part of the Late Pleistocene biota that to a large extent went extinct at the end of Pleistocene[7–9]. It is estimated that c. 83% of all genera of megafauna in South America were lost[34], and *occidentalis* seemingly shared this fate. There is only one Holocene find of *occidentalis*–a single specimen identified among the c. 9000 bird bones collected at a Paleo-Indian settlement in Oregon dating to c. 8000–9000 B.P.[35,36]. In the absence of more paleontological data from the Andean region, it is unknown whether the Late Pleistocene to Early Holocene extinction of *occidentalis* was synchronous throughout its distribution.

Almost all *occidentalis* finds are from the mountainous regions of Mexico and the western U.S., and the fossil record suggests that it must have arrived there before c. 330–250 kya (the age of the

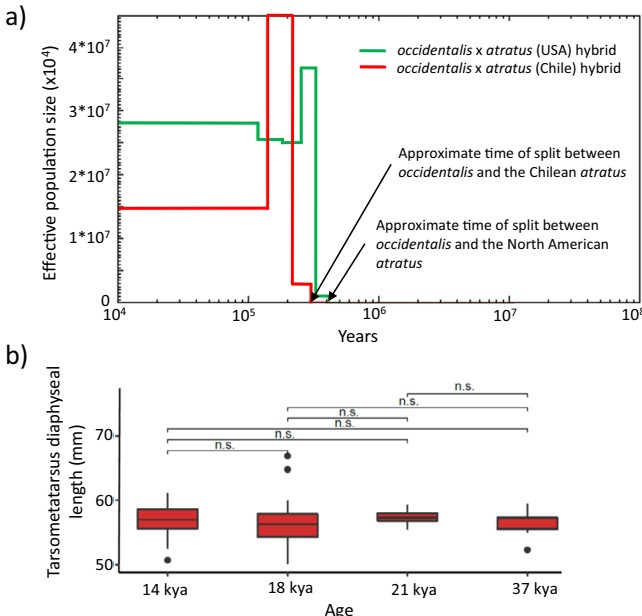

**Fig. 3 Dating of the split between _Coragyps atratus_ and _occidentalis_, and osteometric support for morphological stasis in _occidentalis_ during the Late Pleistocene. a** hPSMC plot based on the artificial F1 hybrid genomes constructed using the fossil _occidentalis_ together with (red) an _atratus_ individual collected in Chile (CA73) and (green) an _atratus_ individual collected in USA (CA03). The time when the hybrid genome indicates an extreme increase in effective population size is inferred to mark the cessation of gene flow between the populations from which the two haploid genomes derive. The hPSMC curves suggest that the ancestors of _occidentalis_ the _atratus_ population in Chile (the one that is phylogenetically closest to _occidentalis_) diverged sometime before 300 kya, while the split between the ancestors of _occidentalis_ and the _atratus_ population in the USA seems to have occurred c. 100 kya earlier. **b** Box-plots of tarsometarsi diaphyseal lengths from _occidentalis_ fossils collected in different tar pits at La Brea, California, indicate that the size of _occidentalis_ has remained constant, despite the dramatic climatic and environmental changes at the end of the Pleistocene.

oldest North American record). As an alternative evolutionary scenario, it could be hypothesized that large-sized _Coragyps_ populations evolved independently in North and South America. However, we find this less likely as direct comparisons of the Peruvian specimens with a large number of _occidentalis_ from the La Brea Tar Pits and from San Josecito Cave in Mexico show complete agreement in morphology and size (Supplementary Table 3 and Note 7). Also, in those parts of the body that can be compared, the fossil have the same body proportions as the North American _occidentalis_ and differs from those of _atratus_ (Supplementary Fig. 4). This indicates that the large-sized _Coragyps_ in North and South America differ from _atratus_ in the same way and thus most likely belong to the same taxon, _occidentalis_. To finally resolve this issue would require a genetic analysis of the North American _occidentalis_ but neither earlier attempts to extract DNA from such specimens (Arthur H. Harris pers. comm.), nor our own attempts (Per Ericson pers. obs.), have been successful so far. At this point, there is nothing to suggest a polyphyletic origin of _occidentalis_-type vultures.

Clearly, the fossil record of _Coragyps_ vultures suggests a sudden evolution of a large-sized highland population with no evidence of intermediate forms between this and its small-sized ancestor. Even though the adaptation to high-elevation conditions must have involved many small steps, it is reasonable to assume that this was a

rapid process because of the exceptionally strong selective pressure caused by cold temperature, low oxygen level, and high UV radiation in the high Andes. We propose that the evolution of _occidentalis_ is an example of "ecological speciation" (a special case of allopatric speciation, which in this case is better termed "ecological diversification" as it makes no taxonomic assumptions). Unlike the gradual divergence between two recently isolated populations that arise predominantly through genetic drift, ecological diversification occurs when one of the isolated populations is exposed to drastically new ecological conditions[37]. Natural selection then becomes the prime driver of differentiation and the strong selection pressure causes rapid genetic and phenotypic change[38–40].

From a geological time perspective, ecological diversification will result in the sudden appearance of new and differently adapted species with no intermediate forms in the fossil record between them and their supposed ancestors. The lack of transitional forms, together with the observation that a species remains largely unchanged until it gets replaced by a new form, is a commonly observed pattern in the fossil record[41–49]. This has led to the hypothesis that evolution often proceeds in a "punctuated" fashion where new species form through bursts of rapid evolution rather than through gradual transformation[42,50,51]. Although it seems clear as the rapid evolution of _occidentalis_ appears as "punctuated" in the fossil record, it is unclear if this was followed by a morphological stasis (equilibrium) after the initial adaptation, as postulated as another criterion for punctuated evolution. Signs of morphological stasis in _occidentalis_ were investigated by comparing size variation between samples collected from different [14]C-dated tar pits in La Brea, California[52]. The analysis shows that _occidentalis_ remained constant in size from 37 to 14 kya (Fig. 3b), a period when the La Brea region experienced dramatic climatic and environmental changes. The last glacial maximum falls within this time interval causing a considerable drop in average temperature, which not least affected the vegetation. Heusser's[53] analysis of pollen data from California shows that the oak and chaparral vegetation around 59 kya changed to a pine, juniper, and cypress woodland around 24 kya. Around 20 kya, the vegetation became a juniper-ponderosa forest, with substantial snow even at such low elevations as La Brea[52]. During the glacial-interglacial transition (14 to 10 kya), the landscape returned to becoming dominated by oak and chaparral, as well as coastal sagebrush. Despite these drastic environmental changes, _occidentalis_ remained constant in size throughout this time (Fig. 3b).

An interesting topic that remains to be investigated is whether the stasis in body size observed in the Late Pleistocene population of _occidentalis_ in La Brea reflects dietary specialization. Changes in body size during the same time in some other scavenging birds (e.g., _Vultur gryphus_ and _Gymnogyps californianus_) occur simultaneously as increased dietary plasticity, which has been suggested to have contributed to the survival of these species across the Late Pleistocene megafaunal extinction[54]. It has been shown that _occidentalis_ in La Brea had a purely terrestrial diet[55], but no similar isotope study has yet been conducted on Late Pleistocene _atratus_ fossils. We do know, however, that the extant _Coragyps atratus_ exhibits great dietary plasticity, feeding on marine resources and occasionally hunting smaller prey. Differences in the dietary plasticity of _atratus_ and _occidentalis_ could potentially be responsible for the different fates of these two taxa by the end of the Pleistocene.

An observation made herein is that morphological change following strong natural selection in a new environment is not necessarily reflected by a general genetic differentiation. As has been shown, the genomic regions targeted by selection may be few and restricted in size, which will have little influence on measurements of the overall genetic differences between taxa[56,57].

The paleogenomic data obtained from a 14,000-year-old fossil of [*Coragyps*] *occidentalis* collected in Andean Peru provides important new information about the evolution of this extinct New World Vulture. First, it clearly shows that the generic affinities of this fossil are with *Coragyps*, thus corroborating previous paleontological identifications. More surprisingly, despite that *occidentalis* is morphologically and morphometrically distinct from the extant Black Vulture *Coragyps atratus*, the paleogenomic analysis does not show it to be genetically differentiated from *atratus*. Instead, it is deeply nested with *atratus*, being genetically most closely related to the geographically close *atratus* populations in southwestern South America. The *occidentalis*-bearing paleontological localities are mostly situated in mountains, indicating that this bird was adapted to high elevations, unlike *atratus*, which is restricted to lowlands. Our genetic results suggest that the initial colonization of high-elevation environments took place in South America c. 300 to 400 kya. The newly established *occidentalis* population was exposed to extreme physiological stress in these novel environments and evolved novel adaptations through "ecological diversification". As this probably happens during a short period of time, not least in a geological time perspective, it leaves few traces in the fossil record and the evolution of a new taxon may appear as sudden ("punctuated"). We believe the evolution of *occidentalis* to be a good example of this, the details of which would not have been discovered without access to paleogenomic information.

## Conclusion

Several Pleistocene species of predators, scavengers, parasites, etc. that had co-evolved with the numerous large herbivores that roamed the Americas at the time, were severely affected by the decline and extinction of megafauna by the end of the epoch. For example, among New World Vultures (Cathartidae), only 5 out of 11 genera survived into the Holocene. In this study, we have combined paleogenomic and paleontological information to study the evolution and extinction of one of these cathartid vultures, [*Coragyps*] *occidentalis*. We found that *occidentalis* evolved after a population of the extant Black Vulture *Coragyps atratus*, which is restricted to lowland regions, colonized the High Andes c. 300 to 400 kya. *occidentalis* was considerably larger and had slightly different body proportions than *atratus*. We argue that the evolution of this now extinct taxon constitutes an example of punctuated evolution resulting from strong natural selection in the extreme, high-altitude environment.

## Materials and methods

**Extraction of ancient and historical DNA, library preparation, and sequencing**.
Laboratory procedures for the ancient DNA (aDNA) Casa del Diablo vulture were carried out in a state-of-the-art facility for aDNA, while the laboratory work for the historical (hDNA) samples from museum study skin were performed independently in another laboratory dedicated to hDNA samples (for a more detailed description of the laboratory procedures for the aDNA work with the Casa del Diablo vulture see Supplementary Methods 1). We obtained ~50 mg bone powder of the Casa del Diablo vulture from the femur (A215) using a Dremel high-speed multi-tool and DNA extraction was done according to the protocol in ref. [58].

Courtesy of the American Museum of Natural History, New York, and the National Museum of Natural History, Smithsonian Institution, Washington D.C., we obtained toe pads of 52 museum study skins of *Coragyps atratus* (Supplementary Table 5). DNA was extracted using the Qiagen QIAamp DNA Mini Kit following the protocol described in ref. [59]. For the double-stranded Illumina library preparation, we followed the protocol of ref. [60] that consists of three steps, blunt-end repair, adapter ligation, and adapter fill-in. For the Casa del Diablo vulture, an additional step of USER enzyme treatment was added before the blunt-end repair (DNA was incubated with 6U USER enzyme for 3 h at 37 °C) to excise uracil residues resulting from post-mortem damage[61,62]. Two size selection/cleaning steps were performed in between the three library preparation steps. Due to its more degraded nature, MinElute spin columns (Qiagen, Hilden, Germany) was used for the Casa del Diablo vulture, while AMPure magnetic beads (Beckman Coulter™) were used for the toepad samples. An analysis of the damage pattern with

mapDamage[63] shows that this procedure efficiently reduced the DNA damage (Supplementary Fig. 5).

Before sequencing, all libraries were amplified and indexed. To control for index hopping, we used dual indexing where both the P5 and the P7 adapters are indexed with unique barcodes[64]. As ligated fragments are randomly amplified in the index-PCR step, a higher number of PCR cycles will increase the clonality (that certain fragments are overrepresented in the final indexed library). To circumvent this problem, we reduced the number of cycles as much as possible (8–12 cycles) and ran multiple independent PCRs (eight for the Casa del Diablo vulture and three for the foot pad samples).

Finally, the Casa del Diablo vulture samples were sequenced on one lane on the Illumina NovaSeq (S4) platform, while the foot pad samples were pooled and sequenced at equal molarity on two lanes on the Illumina NovaSeq (S4) platform.

The Illumina sequencing reads for both the fossil bone and the museum study skins were processed using a custom-designed workflow (github.com/mozesblom) to remove adapter contamination, low-quality bases, and low-complexity reads. Overlapping read pairs were merged using PEAR[65] and PCR duplicates were removed using SuperDeduper[66]. Trimming and adapter removal were done with Trimmomatic v0.32[67] (default settings) and the overall quality and length distribution of sequence reads were inspected with FastQC v0.11.5 (Andrews, http://bioinformatics.babraham.ac.uk/projects/fastqc/) before and after the cleaning. Like for the Casa del Diablo fossil, we treated the museum skin samples with the USER enzyme (as described above) to excise uracil residues resulting from post-mortem damage[61,62].

**DNA extraction, whole-genome sequencing, and de novo assembly of extant *Coragyps atratus***. We assembled a new genome of an adult male *Coragyps atratus* collected at Rio Negro (20º10′S; 58º10′W), Departemento Alto Paraguay, Paraguay, on 15 September 1994 (voucher NRM 947124)[68]. DNA was extracted from EtOH preserved tissue using the KingFisher duo extraction robot and the KingFisher™ Cell and Tissue DNA Kit according to the manufacturer's instructions. Sequencing libraries were generated by Science for Life Laboratory (National Genomics Institute, Stockholm) with the Chromium controller instrument and reagents from 10X Genomics, followed by sequencing on an Illumina HiSeqX sequencer. The de novo assembly and quality control were performed using the nf-core/neutronstar analysis pipeline[69]. The final genome assembly of 130 Gb of high-quality sequence data resulted in 46,819 scaffolds covering 1.2 Gb with a scaffold N50 of 88 kb and a contig N50 of 55 kb. Initial investigations into the gene content of the assembled genome using BUSCOv3 and the Eukaryota BUSCO dataset revealed 75% complete BUSCOs.

*Initial reference mapping, extraction of homologous sequences, and phylogenomic reconstruction*. We used BWA mem v0.7.12[70] to map the trimmed *occidentalis* and *atratus* reads against the *Gymnogyps californianus* genome[71]. We obtained a mapping coverage of 1.25× for *occidentalis* and a mean coverage of 3.3× (range 1.0–8.4×) for the 41 *atratus* individuals sequenced from museum skins. To obtain a large number of sequence homologs of nuclear exonic and intronic loci across the whole genome, we performed searches using profile hidden Markov models (HMM)[72]. Profile HMMs use information from variation in multiple sequence alignments to seek similarities in databases or, as here, genome assemblies[73]. The HMM profiles were based on the alignments of exonic and intronic loci of *Cathartes aura*, *Haliaeetus albicilla* and *Haliaeetus leucocephalus* published by ref. [26]. For each HMM query and taxon, the location on the genome of the most significant hit was identified, and the sequence was parsed out using the genomic coordinates. These steps were carried out using a custom-designed BirdScanner pipeline[74] (github.com/Naturhistoriska/birdscanner). We used the BirdScanner pipeline also to obtain exonic loci to reconstruct the higher-level phylogenetic relationships of *Coragyps*, as well as to obtain both exonic and intronic loci for those *atratus* individuals with the highest sequencing coverage that were used to root the population-level phylogeny (see below). The extracted sequences were then aligned with the other individuals in separate files. Poorly aligned sequences were identified based on a calculated distance matrix using OD-Seq (github.com/PeterJehl/OD-Seq) and excluded from further analyses. We also checked the alignments manually to identify those that included non-homologous sequences. We found that no alignments with a 5% or lower proportion of phylogenetically informative sites included non-homologous sequences, and we used this as a conservative cut-off value for deleting potentially problematic alignments from further analyses. Gaps were treated as missing data in all analyses. Phylogenetic relationships were estimated using Bayesian inference in BEAST2 v2.4.8[75], after first having identified the best-fit model of nucleotide substitutions with IQ-TREE (-m TEST)[76]. In the analyses, we ran the chains for 100 million generations, with trees sampled every 1000 generations, with the first 10% (the burn-in phase) excluded from the analyses.

To investigate the molecular support for referring the *occidentalis* fossil to the genus *Coragyps*, we used 892 exons (a total of 506 kb, Supplementary Table 6) of ten species sampled from the clade Accipitrimorphae (sensu ref. [26]) to which the families Accipitridae, Pandionidae, Sagittaridae and Cathartidae belong, and three outgroups from the families Phasianidae, Ptilonorhynchidae, and Corvidae. Genomes of the following taxa were downloaded from GenBank (accession numbers in parentheses): *Gallus gallus* (GCA000002315), *Cathartes aura*

(GCA000699945), *Gyps himalayensis* (GWHBAOP00000000), *Gypaetus barbatus* (GWHBAOQ00000000), *Buteo japonicus* (GCA010312235), *Haliaeetus albicilla* (GCA000691405), *Sagittarus sepentarius* (GCA013399415), *Pandion haliaetus* (GCA013401275), *Gymnogyps californianus* (GCA018139145), *Amblyornis subalaris* (GCA018881555), and *Corvus cornix* (GCA000738735). In addition, we included the de novo genome of *Coragyps atratus* from the present analysis and the genome of *occidentalis* sequenced here. We used BEAST2 v2.4.8[75] with a general time-reversible (GTR) model for nucleotide substitutions with 5 gamma rate categories (empirical base frequencies and free rates), a relaxed log-normal clock, and a birth–death tree prior. The MCMC chain was set to 50 million generations sampling every 1000 generations and we used Tracer v.1.5[77] to assure that an adequate (~200) effective sample sizes (ESS) had been reached for all parameters indicating proper MCMC mixing and convergence.

Population-level relationships within *Coragyps* were estimated from a total of 1179 intronic loci (Supplementary Table 6). Not all introns could be obtained from all individuals, and the alignments thus include, on average, 28 individuals. The concatenated alignment includes a total of 479 kb. The BEAST2 analysis was run with a general time-reversible (GTR) model for nucleotide substitutions with 5 gamma rate categories (empirical base frequencies and free rates), a relaxed log-normal clock, and a birth–death tree prior. We run the MCMC chain and evaluated the results in the same way as described above. In the population-level phylogeny, we used CA15 from Panama as an outgroup, based on the result of an analysis specifically aimed at finding the root of the *Coragyps atratus* tree (Supplementary Fig. 3). In that analysis, we used alignments of 518 exons (357 kb) and 3633 introns (610 kb) (Supplementary Table 6) obtained for the twelve individuals with high mapping coverage representing the geographic range of *Coragyps atratus*. The higher sequencing coverage in these individuals decreased the amount of missing data and allowed us to align them to homologous sequences obtained for an individual of *Cathartes aura*, a representative of the sister genus of *Coragyps*[30]. Also this dataset was analyzed using BEAST2 with a general time-reversible (GTR) model for nucleotide substitutions with 5 gamma rate categories (empirical base frequencies and free rates), a relaxed log-normal clock, and a birth–death tree prior. The run of the MCMC chain and the evaluation of the results were done as described above.

**Genotype likelihood estimates**. The low sequencing coverage herein makes estimates of genotype likelihoods that incorporate information regarding alignment or assembly uncertainty and base-calling uncertainty a better methodological choice than calling genotypes[78]. We estimated genotype likelihoods (applying the GATK genotype likelihood model) for *occidentalis* and 52 individuals of *atratus* using ANGSD v0.933[79,80], with the following parameters: "-GL 2 -doGlf 2 -doMajorMinor 1 -doMaf 1 -doCounts 1 -minMapQ 15 -minQ 5 -setMinDepthInd 10 -setMaxDepthInd 100 -skipTriallelic 1 -uniqueOnly 1 -only_proper_pairs 0 -SNP_pval 1e-6 -minMaf 0.10 -minInd 10". We used the genotype likelihoods to run NGSadmix[81] and PCAngsd v0.985[82]. In NGSadmix we let K range from 2 to 8, ran 30 iterations for each K and determined the "best" K using a customized R script (https://baylab.github.io/MarineGenomics/week-9-population-structure-using-ngsadmix.html) that utilizes the Cluster Markov Packager Across K (CLUMPAK, http://clumpak.tau.ac.il/index.html). We also used PCAngsd to generate a covariance matrix from the genotype likelihoods estimated in ANGSD. The *eigen()* function in R (R: The R Project for Statistical Computing) was used for the principal component analysis (PCA) and we plotted the samples on the first two principal components.

**Inferring the divergence time between *occidentalis* and *atratus***. We estimated the timing of the split between *occidentalis* and *atratus* using three different methods. First, we translated the genetic distance observed between *occidentalis* and its nearest *atratus* relative (CA73 collected in Chile) in cytochrome *b* into years by using a mean average rate of divergence between two taxa of 2.1% per million years[83]. We did this also for the distance observed between *occidentalis* and CA73 for the concatenation of nuclear introns (485 kb) using a mean average rate of divergence between two taxa of 0.128% per million years[84]. Our third method to date the split between *occidentalis* and *atratus* was to use hPSMC[85] to estimate the time for the end of gene flow between these populations. We created an artificial F1 hybrid genome from pseudo-haploid sequences of *occidentalis* and *atratus*. The method uses the fact that the genomes of the artificial F1 hybrid cannot coalesce more recently than the time of speciation of the two parental populations. This allows hPSMC thus to infer the time since divergence as a measure of cessation of gene flow, and of the effective population size prior to divergence[85,86]. Haploid consensus genomes were reconstructed for the five longest scaffolds from the bam files of *occidentalis* and the phylogenetically closest *atratus* individual (CA73) using samtools mpileup[87] and pu2fa (https://github.com/Paleogenomics/Chrom-Compare). We applied a minimum base quality and mapping quality of 30 and a minimum depth of 5. The sequences were haploidized by randomly picking one base for each position using pu2fa (https://github.com/Paleogenomics/Chrom-Compare). The two haploid fasta files were then combined into a diploid, "F1 hybrid" sequence using the hPSMC tool psmcfa_from_2_fastas.py. After this we ran the psmcfa output through PSMC[88] with the parameters (-p) "4 +30*2 + 4 + 6 + 10"[89], number of iterations = 30, maximum 2N0 coalescent time = 15, initial theta/rho ratio = 5. We used psmc_plot.pl to plot the result, assuming a mutation rate of $1.4 \times 10^{-9}$ per base pair per generation[29] and a generation time of 14.2 years[90].

**Reporting summary**. Further information on research design is available in the Nature Research Reporting Summary linked to this article.

## Data availability
Raw sequence data are available for download from the NCBI Sequence Read Archive (SRA, BioProject PRJNA833756). Phylogenomic data and genotype likelihoods are available from the Dryad Digital Repository (https://doi.org/10.5061/dryad.qz612jmjm)[91].

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

## Acknowledgements

The American Museum of Natural History, New York, and the National Museum of Natural History, Smithsonian Institution, Washington D.C., kindly provided toepad samples of study skins in their collections. We particularly want to thank Paul Sweet and James Dean for arranging these loans. We are grateful for the help and insightful discussions with Virginia Butler, Ken E. Campbell, Fred W. Croxen III, Love Dalén, Pamela Endzweig, Arthur H. Harris, Fritz Hertel, Amadeo Rea, David W. Steadman, and late Storrs L. Olson. Donald R. Prothero kindly provided information about *occidentalis* collected in tar pits at La Brea. The manuscript was improved by suggestions made by Kari Alyssa Prassack and an anonymous reviewer. We want to thank Huishang She for helping with designing several figures. The map in fig. 2c uses data from the eBird Status and Trends Project at the Cornell Lab of Ornithology, eBird.org. We are grateful for this and acknowledge that any opinions, findings, and conclusions or recommendations expressed in this material are those of the authors, and do not necessarily reflect the views of the Cornell Lab of Ornithology. The authors acknowledge support from the National Genomics Infrastructure in Stockholm funded by Science for Life Laboratory, the Knut and Alice Wallenberg Foundation and the Swedish Research Council, and SNIC/Uppsala Multidisciplinary Center for Advanced Computational Science for assistance with massively parallel sequencing and access to the UPPMAX computational infrastructure. This research was funded by the Swedish Research Council (grant 621-2017-3693 to P.G.P.E.) and the National Natural Science Foundation of China (grant NSFC32020103005 to Y.Q.).

## Author contributions

P.G.P.E. and Y.Q. conceptualized the study and acquired funding. M.I., D.Z., P.L., J.-L.T., and A.G. conducted the DNA extractions. M.I. and P.L. prepared the sequencing libraries. L.W., S.D.E., and J.P.H. contributed resources. P.G.P.E. and Y.Q. conducted the bioinformatics and statistical analyses. P.G.P.E., Y.Q., and J.P.H. visualized the results. P.G.P.E. and Y.Q. wrote the manuscript. All authors contributed to the draft and approved the final version of the manuscript.

## Competing interests

The authors declare no competing interests.
