## [Peer Review File · Communications Biology]

Reviewers' comments:

Reviewer #1 (Remarks to the Author):

Paleogenomic sequencing of the Pleistocene New World vulture *Coragyps occidentalis* supports its descent from *Coragyps atratus*. *Coragyps occidentalis* is inferred to have undergone rapid speciation in response to the environmental constraints of a high elevation environment, dispersed northward into western North America, and maintained apparent stasis in its size, body proportion, and morphology across several major environmental shifts. This is supported by morphological data and the paleontological record.

Molecular phylogenetics has increasingly become a litmus test for interpreting evolutionary relationships and the timing of first appearances in the fossil record, even sometimes initially contradicting it only to be corroborated later in light of new fossil discoveries. It's an important technique and in particular when fossil DNA is preserved allows for a suite of methods that can add validity to our understanding of a taxon's evolutionary relationships.

This is an important paper not only for understanding *Coragyps* systematics, but for considering how niche diversification under extreme conditions can lead to rapid speciation and the broader effect that the late Pleistocene extinction had on megafauna-reliant taxa. It should be of interest to a wide audience, with broader implications for understanding the domino effect of extinction on associated taxa, potentially proving useful for predictive models aimed at understanding the current biodiversity crisis.

The authors do an excellent job of answering both their own postulated questions and those that arose as I read through this manuscript and the data they present are clearly replicable. I found their analysis sound and their distribution maps especially useful, though I would have liked to see the inclusion of site ages to gain a better appreciation for the timing of dispersal across the Americas: Fig.1d but with greater detail. The distribution of *C. atratus* as shown in Figure 1a is suggestive of a coastal migration, though I suppose that could also be a preservation artifact.

That said, it did lead me to one question I would like to see the authors better address. Why might have *C. occidentalis* gone extinct in the highlands while *C. atratus* survived and is extant and widespread today? The authors hint at the megafaunal extinction but this does not explain the survival of other large scavenging birds including the longevity and success of *C. atratus* as a species. In some scavenging birds (e.g., *Vultur gryphus* and *Gymnogyps californianus*) we see an increase in body size along with increased dietary plasticity that likely facilitated their survival across the late Pleistocene megafaunal extinction (Perrig et al 2019). In *C. occidentalis* we see stasis in body size and shape. Did this correlate to a similar stasis in dietary behavior? *Coragyps occidentalis* at La Brea is shown to have had a purely terrestrial diet (Fox-Dobbs et al. 2006) which presumably was the case for those from higher elevations as well, while extant *Coragyps atratus* (and the Californian condor) exhibit greater dietary plasticity, feeding on marine resources and occasionally hunting smaller prey. It would therefore be interesting to look isotopically at Pleistocene *C. atratus* in comparison to La Brea and Andean *C. occidentalis* (and ideally much older specimens) to see if *C. atratus* was already more ecologically diverse in its diet than *C. occidentalis*. This is of course well beyond the scope of this paper, but I think is something that, if discussed, would facilitate future studies.

Grammatical issue with the sentence on line 234, otherwise excellent organization, writing, and editing.

Great paper, excited to see it in print.

Fox-Dobbs, K., Stidham, T.A., Bowen, G.J., Emslie, S.D. and Koch, P.L., 2006. Dietary controls on extinction versus survival among avian megafauna in the late Pleistocene. *Geology*, 34(8), pp.685-688.

Perrig, P.L., Fountain, E.D., Lambertucci, S.A. and Pauli, J.N., 2019. Demography of avian scavengers after Pleistocene megafaunal extinction. *Scientific reports*, 9(1), pp.1-9.

Reviewer #2 (Remarks to the Author):

Ericson and colleagues use genomic and morphometric data to demonstrate that an extinct vulture species (*Coragyps occidentalis*) arose from a South American population of the extant black vulture (*Coragyps atratus*) in an example of punctuated equilibrium/evolution. Overall, this is an excellent study that I very much enjoyed reading — it is rich in data, thought-provoking, and the results are carefully dissected and interpreted (though much of the dissection is necessarily relegated to the SI). It is a great example of how palaeogenomic data can complement more traditional methods to develop novel insights. The only real concerns I have with this manuscript that prevent me from outright recommending acceptance in its current form relate to the BEAST analyses and phylogeographic interpretation (see below) — my other comments mainly relate to clarity and consistency. If the authors are able to address all these points I would be happy to recommend publication (and I would be pleased to review a revised version of the manuscript).

Section: Results and Discussion - Paleogenomic perspectives on [*Coragyps*] *occidentalis*

Please state depth of coverage of the genomic data in the first paragraph of this section (otherwise the reader needs to skip ahead to lines 310-311 to find this info, which is relevant to the interpretation of results and strength of conclusion).

Lines 134-137: "The results indicate that *Coragyps atratus* originated in the Panamanian biogeographical area, which includes Central America and northern South America (Fig 2c), from where *atratus* spread both north to the Nearctic region via the Mexican biogeographical area, and south to the Amazonian and, subsequently, the Patagonian-Andean biogeographical areas."

While this might be true, I do not believe that the authors' results "indicate" that it is the case (especially in the absence of any explicit phylogeographic model testing). Their favoured scenario invokes two episodes of dispersal: one from Central America northward (the ancestor of clade 2 in Fig2C) and one from Central America southward (the ancestor of clade 4+5+6 in Fig2C). An equally parsimonious explanation (based on the phylogeny alone) is that *atratus* originated in South America, and dispersed twice to Central and/or North America (the ancestors of clade 1 and clade 2+3). In my opinion, the authors should better substantiate their favoured scenario (perhaps with fossil occurrence data), soften the wording (e.g. "consistent with" rather than "indicate"), or just remove this entirely (since it doesn't matter for their main conclusions).

Lines 193-197: "As an alternative evolutionary scenario, it could be hypothesized that large-sized *Coragyps* populations evolved independently in North and South America. However, we find this less likely as direct comparisons of the Peruvian specimens with a large number of *occidentalis* from the La Brea Tar Pits and from San Josecito Cave in Mexico show complete agreement in morphology and size (Supplementary Fig. 3, Table 3 and Text 9)."

I was very pleased to see the authors explicitly mention this alternative hypothesis, because I found myself wondering about the origins of North American *occidentalis* as I read the preceding text. I am convinced by the authors' argument, but only after reading their SI text 4 where they dissect (and reject) the possibility that *occidentalis* is an "upscaled" *atratus*, with allometric scaling explaining much of the difference between the two taxa. I strongly recommend that the authors add a new sentence (and reference to Supplementary Text 4) to this paragraph in the main text drawing attention to this specific point.

Lines 292-295 "As erroneous DNA degradation patterns almost exclusively appear at the ends of sequence reads, we arbitrarily shortened all reads obtained from the museum study skins by deleting 5bp from both ends in order to reduce this "noise". An analysis of the remaining damage pattern with mapDamage shows that this procedure efficiently reduced the remaining DNA damage (Supplementary Fig. 4)."

The fossil bone sample was treated using the USER enzyme mix, which should have removed practically all evidence of post-mortem damage. Presumably this is why the reads from the occidentalis bone were not subjected to the 5bp trimming that was applied to the data from the museum skins. But — according to the legend — Figure S4 is generated from the occidentalis bone data, so it does not show the effects of the trimming as the text implies. Perhaps the authors could add also add a before- and after-trimming plot for one of the museum skins?

Line 322: "...root the phylogeography..." (and again on line 351 and elsewhere)

Phylogeography is a field of study and cannot be used as a singular noun. The authors are here referring to a phylogeny (from which they subsequently draw conclusions about phylogeographic structure). No spatial data are actually used in the analysis. I strongly recommend the authors change their usage of "phylogeography" to "intraspecies phylogeny" or "population-level phylogeny" to differentiate it from their earlier higher-level phylogenetic analysis.

Line 326: "...indicated by an extreme proportion of variable positions in the alignment..."

Was an objective and consistent cut-off used to perform this filtering? Some more detail would be useful.

Line 327: "...[removed sites] that contained no phylogenetic information (no parsimony-informative sites)."

Autapomorphic sites (i.e. singletons) are parsimony-uninformative but are informative for estimating terminal branch lengths (and substitution rate parameters) in Bayesian analyses. I don't think excluding them in this case affects the authors' interpretation of their results (they do not consider branch lengths of their nuclear phylogenies), but I thought I should highlight this point for their future reference.

Line 342-344: "A generalized time reversible (GTR) model for nucleotide substitutions, with empirical base frequencies and free rates, was used in the Beast2 analysis."

The authors should also state what clock model and tree prior they used for all BEAST2 analyses (also note that the more widely used spelling is "BEAST2" not "Beast2"), as well as how convergence of the MCMC was assessed.

Line 345-354: BEAST2 analyses

It is important to note that BEAST2 infers a rooted ultrametric phylogeny, which should never be re-rooted post hoc (because the posterior support values and branch lengths are co-estimated alongside that root position). If there is prior information about the root position, then this must be specified at the beginning of the analysis by constraining the monophyly of one or more clades a priori. The order in which the methods are presented here implies that the authors have first conducted an analysis with a large number of atratus individuals, but then a subsequent analysis with a reduced set of individuals to root the results of the former (which would be improper, because the results of the former are already probabilistically rooted). This is an important methodological point that potentially affects the topology and interpretation of the phylogeny and must be clarified (and repeated correctly if necessary).

Line 378-379: "We created an artificial F1 hybrid genome from pseudo-haploid sequences of occidentalis and atratus."

What method was used to generate pseudohaploid genotypes and assemble them into artificial F1 genomes? Is this functionality built into hPSMC?

Supplementary Text 4 line 138: I think "stratus" should be "atratus"

Response to the reviewers

Reviewer #1 (Remarks to the Author):

Paleogenomic sequencing of the Pleistocene New World vulture *Coragyps occidentalis* supports its descent from *Coragyps atratus*. *Coragyps occidentalis* is inferred to have undergone rapid speciation in response to the environmental constraints of a high elevation environment, dispersed northward into western North America, and maintained apparent stasis in its size, body proportion, and morphology across several major environmental shifts. This is supported by morphological data and the paleontological record.

Molecular phylogenetics has increasingly become a litmus test for interpreting evolutionary relationships and the timing of first appearances in the fossil record, even sometimes initially contradicting it only to be corroborated later in light of new fossil discoveries. It's an important technique and in particular when fossil DNA is preserved allows for a suite of methods that can add validity to our understanding of a taxon's evolutionary relationships.

This is an important paper not only for understanding *Coragyps* systematics, but for considering how niche diversification under extreme conditions can lead to rapid speciation and the broader effect that the late Pleistocene extinction had on megafauna-reliant taxa. It should be of interest to a wide audience, with broader implications for understanding the domino effect of extinction on associated taxa, potentially proving useful for predictive models aimed at understanding the current biodiversity crisis.

Q1. The authors do an excellent job of answering both their own postulated questions and those that arose as I read through this manuscript and the data they present are clearly replicable. I found their analysis sound and their distribution maps especially useful, though I would have liked to see the inclusion of site ages to gain a better appreciation for the timing of dispersal across the Americas: Fig. 1d but with greater detail. The distribution of *C. atratus* as shown in Figure 1a is suggestive of a coastal migration, though I suppose that could also be a preservation artifact.

RESPONSE: Thanks for suggesting this interesting idea to explore if the temporal distribution of the sites may provide information on the timing of dispersal of *Coragyps atratus*. We have made several tries with numbering the sites in Fig. 1a but the figure has become much too complicated to be readable because of the accumulation of samples, especially in Florida. Instead we have included the geographic locations for the oldest finds of *atratus* in Fig 1d (see below). From that it is clear that no inference can be made about the timing of dispersals across the Americas based on the geographic location of these samples. We have added a couple of sentences about this to the text (line 170 in the revision with track-changes):

“The temporal and geographic distributions of the *atratus* fossils provide, however, no information to the timing of the dispersals or what routes they followed. Almost all of the oldest finds of *atratus* have been made in Florida (Fig 1d), which most likely can be explained by differences in preservation and paleontological activity across its distribution area.”

Q2. That said, it did lead me to one question I would like to see the authors better address. Why might have *C. occidentalis* gone extinct in the highlands while *C. atratus* survived and is extant and widespread today? The authors hint at the megafaunal extinction but this does not explain the survival of other large scavenging birds including the longevity and success of *C. atratus* as a species. In some scavenging birds (e.g., *Vultur gryphus* and *Gymnogyps californianus*) we see an increase in body size along with increased dietary plasticity that likely facilitated their survival across the late Pleistocene megafaunal extinction (Perrig et al 2019). In *C. occidentalis* we see stasis in body size and shape. Did this correlate to a similar stasis in dietary behavior? *Coragyps occidentalis* at La Brea is shown to have had a purely terrestrial diet (Fox-Dobbs et al. 2006) which presumably was the case for those from higher elevations as well, while extant *Coragyps atratus* (and the Californian condor) exhibit greater dietary plasticity, feeding on marine resources and occasionally hunting smaller prey. It would therefore be interesting to look isotopically at Pleistocene *C. atratus* in comparison to La Brea and Andean *C. occidentalis* (and ideally much older specimens) to see if *C. atratus* was already more ecologically diverse in its diet than *C. occidentalis*. This is of course well beyond the scope of this paper, but I think is something that, if discussed, would facilitate future studies.

RESPONSE: We thank the reviewer for this very interesting suggestion. The observation of increased body sizes of certain scavenging birds along with increased dietary plasticity clearly raises the question whether the stasis in body size in *occidentalis* reflects dietary specialization. From the little what is known about *occidentalis*' diet, it indeed differs from that of *atratus*, which often inhabits coastal environments. It would definitely be interesting to

explore this in future studies. We have added the following text to the Results and Discussion (line 287):

“An interesting topic that remains to be investigated is whether the stasis in body size observed in the Late Pleistocene population of *occidentalis* in La Brea reflects dietary specialization. Changes in body size during the same time in some scavenging birds (e.g., *Vultur gryphus* and *Gymnogyps californianus*) occur simultaneously as increased dietary plasticity, which has been suggested to have contributed to the survival of these species across the Late Pleistocene megafaunal extinction (Perrig et al. 2019). It has been shown that *occidentalis* in La Brea had a purely terrestrial diet (Fox-Dobbs et al. 2006), but no similar isotope study has yet been conducted on Late Pleistocene *atratus* fossils. We do know, however, that the extant *Coragyps atratus* exhibits a great dietary plasticity, feeding on marine resources and occasionally hunting smaller prey. Differences in their dietary plasticity could potentially be responsible for the different fates of these two taxa by the end of the Pleistocene.”

Q3. Grammatical issue with the sentence on line 234, otherwise excellent organization, writing, and editing.

RESPONSE: We have changed the wording of this sentence to (line 297):

“An observation made herein is that morphological change following strong natural selection in a new environment is not necessarily reflected by a general genetic differentiation.”

Great paper, excited to see it in print.

Fox-Dobbs, K., Stidham, T.A., Bowen, G.J., Emslie, S.D. and Koch, P.L., 2006. Dietary controls on extinction versus survival among avian megafauna in the late Pleistocene. *Geology*, 34(8), pp.685-688.

Perrig, P.L., Fountain, E.D., Lambertucci, S.A. and Pauli, J.N., 2019. Demography of avian scavengers after Pleistocene megafaunal extinction. *Scientific reports*, 9(1), pp.1-9.

Reviewer #2 (Remarks to the Author):

Ericson and colleagues use genomic and morphometric data to demonstrate that an extinct

vulture species (*Coragyps occidentalis*) arose from a South American population of the extant black vulture (*Coragyps atratus*) in an example of punctuated equilibrium/evolution. Overall, this is an excellent study that I very much enjoyed reading — it is rich in data, thought-provoking, and the results are carefully dissected and interpreted (though much of the dissection is necessarily relegated to the SI). It is a great example of how palaeogenomic data can complement more traditional methods to develop novel insights. The only real concerns I have with this manuscript that prevent me from outright recommending acceptance in its current form relate to the BEAST analyses and phylogeographic interpretation (see below) — my other comments mainly relate to clarity and consistency. If the authors are able to address all these points I would be happy to recommend publication (and I would be pleased to review a revised version of the manuscript).

RESPONSE: Thank you for your encouraging summary and constructive comments to improve our work.

Section: Results and Discussion - Paleogenomic perspectives on [*Coragyps*] *occidentalis*

Q1. Please state depth of coverage of the genomic data in the first paragraph of this section (otherwise the reader needs to skip ahead to lines 310-311 to find this info, which is relevant to the interpretation of results and strength of conclusion).

RESPONSE: Thanks for suggesting this. We have moved the information of the mapping coverage of the paleogenome of *occidentalis* to the first paragraph of the subsection: *Paleogenomic perspectives on [Coragyps] occidentalis*, in line 136 as below.

“In order to ascertain the generic affinity of *occidentalis*, we extracted DNA from the Casa del Diablo specimens and used whole-genome resequencing to generate a paleogenome of *occidentalis* with a mapping coverage of 1.25×”

Q2. Lines 134-137: "The results indicate that *Coragyps atratus* originated in the Panamanian biogeographical area, which includes Central America and northern South America (Fig 2c), from where *atratus* spread both north to the Nearctic region via the Mexican biogeographical area, and south to the Amazonian and, subsequently, the Patagonian-Andean biogeographical areas."

While this might be true, I do not believe that the authors' results "indicate" that it is the case (especially in the absence of any explicit phylogeographic model testing). Their favoured scenario invokes two episodes of dispersal: one from Central America northward (the ancestor of clade 2 in Fig2C) and one from Central America southward (the ancestor of clade 4+5+6 in Fig2C). An equally parsimonious explanation (based on the phylogeny alone) is that *atratus* originated in South America, and dispersed twice to Central and/or North America (the ancestors of clade 1 and clade 2+3). In my opinion, the authors should better substantiate their favoured scenario (perhaps with fossil occurrence data), soften the wording (e.g. "consistent

with" rather than "indicate"), or just remove this entirely (since it doesn't matter for their main conclusions).

RESPONSE: Thanks for pointing this out. In the revised ms we have inferred the ancestral area for the basal node within the population-level phylogeny for *Coragyps atratus*. We rooted the tree with an individual from Panama based on the analysis with high coverage individuals and the outgroup *Cathartes aura* (see Q9 below). The idea of this analysis was to include also exonic regions that were possible to align with the sister taxon of *Coragyps*. Another individual from Panama falls basal within the *atratus* ingroup in population-level analysis. We thus infer that there is a 50% probability that the ancestral area of the entire *Coragyps* tree is the Panamanian biogeographical area, while there is a 25% probability each for a North American and South American origin, respectively. Although neither of the latter hypotheses can be excluded, they require more complex dispersal scenarios than does an origin in Central America. The revised text is (line 156):

“To further clarify the geographic origin of *occidentalis*, we conducted a phylogeographic analysis of 1,179 nuclear introns (totaling 586 kb) obtained from 42 recent *atratus* individuals (collected throughout the geographical distribution of *atratus*) and the Casa del Diablo specimen of *occidentalis*, using Bayesian inference. The phylogeny was rooted with one of the samples from Panama (CA15) based on a previous analysis of smaller number of individuals with higher coverage. The higher coverage allowed us to include exonic regions for which also *Cathartes aura*²⁹, a representative of the sister clade to *Coragyps*³⁰, could be aligned (Supplementary Fig. 3). In the population-level phylogeny the *Coragyps* samples shows a distinct geographic clustering (Fig. 2c). From this it can be inferred that the ancestral area of *Coragyps atratus* with 50% probability is in the Panama region, compared to a 25% probability each for a North American or a South American origin (Fig. 2d). If the species evolved in the Panamanian biogeographical area (that includes Central America and northern South America) it may then have spread both north to the Nearctic region via the Mexican biogeographical area, and south to the Amazonian and, subsequently, the Patagonian-Andean biogeographical areas. Such a dispersal scenario also seems more logical than what would follow from an origin of the species in either North or South America.”

Q3. Lines 193-197: "As an alternative evolutionary scenario, it could be hypothesized that large-sized *Coragyps* populations evolved independently in North and South America. However, we find this less likely as direct comparisons of the Peruvian specimens with a large number of *occidentalis* from the La Brea Tar Pits and from San Josecito Cave in Mexico show complete agreement in morphology and size (Supplementary Fig. 3, Table 3 and Text 9)."

I was very pleased to see the authors explicitly mention this alternative hypothesis, because I found myself wondering about the origins of North American *occidentalis* as I read the preceding text. I am convinced by the authors' argument, but only after reading their SI text 4

where they dissect (and reject) the possibility that *occidentalis* is an "upscaled" *atratus*, with allometric scaling explaining much of the difference between the two taxa. I strongly recommend that the authors add a new sentence (and reference to Supplementary Text 4) to this paragraph in the main text drawing attention to this specific point.

RESPONSE: We thank for this suggestion and we have added to the text the following to clarify our opinion about the taxonomic status of the South and North American large-sized *Coragyps*:

“However, we find this less likely as direct comparisons of the Peruvian specimens with a large number of *occidentalis* from the La Brea Tar Pits and from San Josecito Cave in Mexico show complete agreement in morphology and size (Supplementary Table 3 and Note 7). Also, in those parts of the body that can be compared, the fossil have the same body proportions as the North American *occidentalis* and differs from those of *atratus* (Supplementary Fig. 3). This indicates that the large-sized *Coragyps* in North and South America differ from *atratus* in the same way and thus most likely belong to the same taxon, *occidentalis*.”

Q4. Lines 292-295 "As erroneous DNA degradation patterns almost exclusively appear at the ends of sequence reads, we arbitrarily shortened all reads obtained from the museum study skins by deleting 5bp from both ends in order to reduce this “noise”. An analysis of the remaining damage pattern with mapDamage shows that this procedure efficiently reduced the remaining DNA damage (Supplementary Fig. 4)."

The fossil bone sample was treated using the USER enzyme mix, which should have removed practically all evidence of post-mortem damage. Presumably this is why the reads from the *occidentalis* bone were not subjected to the 5bp trimming that was applied to the data from the museum skins. But — according to the legend — Figure S4 is generated from the *occidentalis* bone data, so it does not show the effects of the trimming as the text implies. Perhaps the authors could add also add a before- and after-trimming plot for one of the museum skins?

RESPONSE: Thanks to this comment by the reviewer we have discovered an error in our description of the method used to reduce the influence by sequence damage. In fact, not only the fossil reads, but also the reads from the study skins were USER treated. The trimming of 5bp that we have used previously for museum study skins was not used here. The figure below shows the results of FastQC analysis of the reads from a *Coragyps atratus* skin collected in 1895 for which the sequencing libraries were treated with the USER enzyme mix. We have changed the text in the ms (line 355):

“Like for the Casa del Diablo fossil, we treated the museum skin samples with the USER enzyme (as described above) to excise uracil residues resulting from post-mortem damage.”

Q5. Line 322: "...root the phylogeography..." (and again on line 351 and elsewhere)

Phylogeography is a field of study and cannot be used as a singular noun. The authors are here referring to a phylogeny (from which they subsequently draw conclusions about phylogeographic structure). No spatial data are actually used in the analysis. I strongly recommend the authors change their usage of "phylogeography" to "intraspecies phylogeny" or "population-level phylogeny" to differentiate it from their earlier higher-level phylogenetic analysis.

RESPONSE: Thanks for pointing this out! We have chosen to change "phylogeography" to "population-level phylogeny".

Q6. Line 326: "...indicated by an extreme proportion of variable positions in the alignment..."

Was an objective and consistent cut-off used to perform this filtering? Some more detail would be useful.

RESPONSE: We noted that even after using OD-Seq to identify and remove poor intron alignment some bad alignments remained, which were indicated by a high proportion of "phylogenetically informative sites". Through manual checking of the alignments we found that turned out that using a conservative 5% cut-off would ensure that no bad alignment remained. After applying this cut-off, we got 1,179 intron alignments (586 kb) available for the phylogenetic analyses. We have added an explanation of this to Methods:

"We found that no alignments with 5% or lower proportion of phylogenetically informative sites included non-homologous sequences, and we used this as a conservative cut-off value for deleting potentially problematic alignments from the further analyses."

Q7. Line 327: "...[removed sites] that contained no phylogenetic information (no parsimony-informative sites)."

Autapomorphic sites (i.e. singletons) are parsimony-uninformative but are informative for estimating terminal branch lengths (and substitution rate parameters) in Bayesian analyses. I

don't think excluding them in this case affects the authors' interpretation of their results (they do not consider branch lengths of their nuclear phylogenies), but I thought I should highlight this point for their future reference.

RESPONSE: We thank for this important comment. It is true we don't make use of branch lengths in this study, but we will keep it in mind for the future.

Q8. Line 342-344: "A generalized time reversible (GTR) model for nucleotide substitutions, with empirical base frequencies and free rates, was used in the Beast2 analysis."

The authors should also state what clock model and tree prior they used for all BEAST2 analyses (also note that the more widely used spelling is "BEAST2" not "Beast2"), as well as how convergence of the MCMC was assessed.

RESPONSE: We have added more information about the BEAST analyses. We have replaced the three paragraphs between lines 398 and 431 with the following text:

"To investigate the molecular support for referring the occidentalis fossil to the genus *Coragyps*, we used 892 exons (a total of 506 kb, Supplementary Table 6) of ten species sampled from the clade Accipitrimorphae (sensu ref.²⁶) to which the families Accipitridae, Pandionidae, Sagittaridae and Cathartidae belong, and three outgroups from the families Phasianidae, Ptilonorhynchidae, and Corvidae. Genomes of the following taxa were downloaded from GenBank (accession numbers in parentheses): *Gallus gallus* (GCA000002315), *Cathartes aura* (GCA000699945), *Gyps himalayensis* (GWHBAOP000000000), *Gypaetus barbatus* (GWHBAOQ000000000), *Buteo japonicus* (GCA010312235), *Haliaeetus albicilla* (GCA000691405), *Sagittarus sepentarius* (GCA013399415), *Pandion haliaetus* (GCA013401275), *Gymnogyps californianus* (GCA018139145), *Amblyornis subalaris* (GCA018881555), and *Corvus cornix* (GCA000738735). In addition, we included the de novo genome of *Coragyps atratus* from the present analysis and the genome of occidentalis sequenced here. We used BEAST2 v2.4.8⁷⁵ with a general time-reversible (GTR) model for nucleotide substitutions with 5 gamma rate categories (empirical base frequencies and free rates), a relaxed log-normal clock, and a birth–death tree prior. The MCMC chain was set to 50 million generations sampling every 1,000 generations and we used Tracer v.1.5⁷⁷ to assure that an adequate (~200) effective sample sizes (ESS) had been reached for all parameters indicating proper MCMC mixing and convergence.

Population-level relationships within *Coragyps* were estimated from a total of 1,179 intronic loci (Supplementary Table 6). Not all introns could be obtained from all individuals and the alignments thus include on average 28 individuals. The concatenated alignment includes a total of 479 kb. The BEAST2 analysis was run with a general time-reversible (GTR) model for nucleotide substitutions with 5 gamma rate categories (empirical base frequencies and free rates), a relaxed log-normal clock, and a birth–death tree prior. We run the MCMC chain and evaluated the results in the same way as described above. In the population-level phylogeny

we used CA15 from Panama as outgroup, based on the result of an analysis specifically aimed at finding the root of the *Coragyps atratus* tree (Supplementary Fig. 3). In that analysis we used alignments of 518 exons (357 kb) and 3,633 introns (610 kb) (Supplementary Table 6) obtained for the twelve individuals with high mapping coverage representing the geographic range of *Coragyps atratus*. The higher sequencing coverage in these individuals decreased the amount of missing data and allowed us to align them to homologous sequences obtained for an individual of *Cathartes aura*, a representative of the sister genus of *Coragyps*³⁰. Also this data set was analyzed using BEAST2 with a general time-reversible (GTR) model for nucleotide substitutions with 5 gamma rate categories (empirical base frequencies and free rates), a relaxed log-normal clock, and a birth–death tree prior. We applied the same sequencing strategy as described above.”

Q9. Line 345-354: BEAST2 analyses

It is important to note that BEAST2 infers a rooted ultrametric phylogeny, which should never be re-rooted post hoc (because the posterior support values and branch lengths are co-estimated alongside that root position). If there is prior information about the root position, then this must be specified at the beginning of the analysis by constraining the monophyly of one or more clades a priori. The order in which the methods are presented here implies that the authors have first conducted an analysis with a large number of *atratus* individuals, but then a subsequent analysis with a reduced set of individuals to root the results of the former (which would be improper, because the results of the former are already probabilistically rooted). This is an important methodological point that potentially affects the topology and interpretation of the phylogeny and must be clarified (and repeated correctly if necessary).
RESPONSE: Admittedly, we were not aware of these effects when re-rooting a BEAST tree. We are grateful to learn this. Consequently, we have re-run the population-phylogeny analysis using the sample CA15 as outgroup. The reason for using this sample as outgroup is that it was recovered as basal in the previous BEAST analysis that also included *Cathartes aura*, the sistergroup of *Coragyps*. *Cathartes* was not possible to align for all introns we used in the population-level analysis so we conducted a separate analysis in which we only used the *atratus* individuals with high coverage for which we also could extract exons, as these probably align more reliably with *Cathartes*. This analysis was described in the text but now we also added the resulting tree of this analysis (see below) to the Supplementary material as Supplementary Fig. 3). The new population-level phylogeny with CA15 as outgroup is almost identical with the previous and the places where they differ do not lead to any changes of the interpretations. Below we show the result of the analysis aimed at finding the root of the *Coragyps atratus* tree (top), and the new population-level phylogeny using CA15 as outgroup (bottom). Note that another individual from Panama falls basal within the ingroup.

Q10. Line 378-379: "We created an artificial F1 hybrid genome from pseudo-haploid sequences of occidentalis and atratus."

What method was used to generate pseudohaploid genotypes and assemble them into artificial F1 genomes? Is this functionality built into hPSMC?

RESPONSE: Thanks for noticing this. The haploidization is integrated in the workflow for

hPSMC, but we should acknowledge the method used. We thus have changed the sentence starting on line 458 to:

“Haploid consensus genomes were reconstructed for the five longest scaffolds from the bam files of occidentalis and the phylogenetically closest atratus individual (CA73) using samtools mpileup85 and pu2fa (<https://github.com/Paleogenomics/Chrom-Compare>).”

Q11. Supplementary Text 4 line 138: I think "stratus" should be "atratus"

RESPONSE: Changed.

REVIEWERS' COMMENTS:

Reviewer #1 (Remarks to the Author):

The authors have fully addressed my questions and concerns. I also appreciated their rephrased description on the inferred Panamanian biogeographical ancestral origin for *Coragyps atratus*. I have no further comments or concerns.

Reviewer #2 (Remarks to the Author):

Ericson and colleagues have addressed all of the concerns I had with the original submission. I have only one minor comment on this revised version: the method used for the ancestral area reconstruction is not clearly explained. I assume that it was part of the BEAST analysis (with the distribution encoded as a discrete trait) but that is not explicitly stated in the manuscript. The result of the analysis itself looks fine though, and the interpretation in the Discussion has been updated appropriately, so I have no concerns there. I am happy to recommend that this manuscript be accepted for publication. Congratulations to the authors, and I look forward to seeing the article in print.

Response to the reviewers

Reviewer #1 (Remarks to the Author):

The authors have fully addressed my questions and concerns. I also appreciated their rephrased description on the inferred Panamanian biogeographical ancestral origin for *Coragyps atratus*. I have no further comments or concerns.

RESPONSE: We thank the reviewer for the help to improve our ms.

Reviewer #2 (Remarks to the Author):

Ericson and colleagues have addressed all of the concerns I had with the original submission. I have only one minor comment on this revised version: the method used for the ancestral area reconstruction is not clearly explained. I assume that it was part of the BEAST analysis (with the distribution encoded as a discrete trait) but that is not explicitly stated in the manuscript. The result of the analysis itself looks fine though, and the interpretation in the Discussion has been updated appropriately, so I have no concerns there. I am happy to recommend that this manuscript be accepted for publication. Congratulations to the authors, and I look forward to seeing the article in print.

RESPONSE: We thank the reviewer for the help to improve our ms. We did not conduct a computerized ancestral areas analysis as the states of the basal nodes are unequivocal assessed also without this. The node for which we inferred the ancestral areas consists of two daughter-lineages. One includes the Central American taxa, while the other is split into two clades – one North American and one South American. The ancestral states in node of interest then becomes 50% Central America, 25% North America, and 25% South America. We than the reviewer to point this out in a previous comment, which gave us the opportunity to correct our mistake.